# Identification of Candidate Lung Function-Related Plasma Proteins to Pinpoint Drug Targets for Common Pulmonary Diseases: A Comprehensive Multi-Omics Integration Analysis

**DOI:** 10.3390/cimb47030167

**Published:** 2025-03-01

**Authors:** Yansong Zhao, Lujia Shen, Ran Yan, Lu Liu, Ping Guo, Shuai Liu, Yingxuan Chen, Zhongshang Yuan, Weiming Gong, Jiadong Ji

**Affiliations:** 1Department of Biostatistics, School of Public Health, Cheeloo College of Medicine, Shandong University, Wenhua West Road, Jinan 250012, China; zhaoyansong@mail.sdu.edu.cn (Y.Z.); shenlujia@mail.sdu.edu.cn (L.S.); yanran_2020@mail.sdu.edu.cn (R.Y.); guopingsph@mail.sdu.edu.cn (P.G.); 202136464@mail.sdu.edu.cn (S.L.); chenyingxuan@mail.sdu.edu.cn (Y.C.); yuanzhongshang@sdu.edu.cn (Z.Y.); 2Institute for Medical Dataology, Shandong University, 12550, Erhuan East Road, Jinan 250003, China; 3Department of Biostatistics, University of Michigan, Ann Arbor, MI 48109, USA; luliuu@umich.edu; 4Center for Statistical Genetics, University of Michigan, Ann Arbor, MI 48109, USA; 5Department of Statistics, School of Mathematics, Shandong University, Shanda South Street, Jinan 250100, China

**Keywords:** proteomics, proteome-wide association study, therapeutic, lung function, bioinformatics methods

## Abstract

The genome-wide association studies (GWAS) of lung disease and lung function indices suffer from challenges to be transformed into clinical interventions, due to a lack of knowledge on the molecular mechanism underlying the GWAS associations. A proteome-wide association study (PWAS) was first performed to identify candidate proteins by integrating two independent largest protein quantitative trait loci datasets of plasma proteins and four large-scale GWAS summary statistics of lung function indices (forced expiratory volume in 1 s (FEV1), forced vital capacity (FVC), FEV1/FVC and peak expiratory flow (PEF)), followed by enrichment analysis to reveal the underlying biological processes and pathways. Then, with a discovery dataset, we conducted Mendelian randomization (MR) and Bayesian colocalization analyses to select potentially causal proteins, followed by a replicated MR analysis with an independent dataset. Mediation analysis was also performed to explore the possible mediating role of these indices on the association between proteins and two common lung diseases (chronic obstructive pulmonary disease, COPD and Asthma). We finally prioritized the potential drug targets. A total of 210 protein–lung function index associations were identified by PWAS, and were significantly enriched in the pulmonary fibrosis and lung tissue repair. Subsequent MR and colocalization analysis identified 59 causal protein-index pairs, among which 42 pairs were replicated. Further mediation analysis identified 3 potential pathways from proteins to COPD or asthma mediated by FEV1/FVC. The mediated proportion ranges from 68.4% to 82.7%. Notably, 24 proteins were reported as druggable targets in Drug Gene Interaction Database, among which 8 were reported to interact with drugs, including *FKBP4*, *GM2A*, *COL6A3*, *MAPK3*, *SERPING1*, *XPNPEP1*, *DNER*, and *FER*. Our study identified the crucial plasma proteins causally associated with lung functions and highlighted potential mediating mechanism underlying the effect of proteins on common lung diseases. These findings may have an important insight into pathogenesis and possible future therapies of lung disorders.

## 1. Introduction

Impaired lung function could promote the risk of various diseases [1,2,3,4] and has been proved as a crucial predictor of mortality [5,6]. Many lung diseases are difficult to detect in the early stages and are often under-diagnosed without spirometry [7]. Indeed, the treatment and monitoring of many chronic respiratory diseases is a gradual and time-consuming process [8,9]. Therefore, the corresponding non-invasive spirometry assessment, such as forced expiratory volume in 1 s (FEV_1_) and forced vital capacity (FVC), provides the opportunity to investigate lung functions and pulmonary diseases more effectively and conveniently, as these indices can reflect the status of lung function and are easily to be measured and obtained. Genome-wide association studies (GWAS) have identified thousands of variants and revealed the significant single nucleotide polymorphisms (SNPs) for lung diseases along with lung function indices [10,11]. However, it remains challenging to utilize these GWAS associations to achieve in-depth interventional design in clinical practice, due to the lack of knowledge on the underlying molecular mechanism of the GWAS associations.

One promising approach to facilitate the interpretation on how the GWAS risk variants contribute to the outcome traits is to integrate GWAS with functional genomics. Among sorts of functional molecules, proteins always undertake the major function of biological processes and are likely to be the alternative and preferable drug targets [12,13]. Indeed, plasma proteins, especially those involved in immune or inflammatory responses, have been suggested to contribute to lung functions and related disorders in several investigations [14,15,16,17]. Although these studies have reported the associations between plasma proteins and lung functions, the scales of proteins included in these analyses are limited compared to whole plasma proteome.

More proteins should be involved to facilitate the exploration of more protein–lung function associations. In addition, these studies failed to tease out the relationship between the proteins, lung function indices and common lung disease.

Several statistical genetics methods have been developed to detect the potential trait-related proteins. A proteome-wide association Study (PWAS) is proposed to identify the proteins associated with disease or disease-related traits [18]. Mendelian randomization (MR) analysis has become an efficient tool to identify the causal relationship between proteins and disease, while colocalization (COLOC) analysis is used to identify the shared causal variants between proteins and diseases [19]. Mediation analysis [20] is always carried out to figure out the potential mediating mechanism. All these analyses could, to some extent, identify the potential proteins that are causally associated with complex disease and benefit the proteomics-driven drug target discovery, thus facilitating initial drug development. Recently, more attentions have been paid on investigating the proteins associated with lung functions. However, these limited studies either lack independent replication analysis [15] or lack comprehensive analysis [21], which would make the findings not robust against different dataset or different analysis methods.

In this study, we integrated two independent, publicly available largest protein quantitative trait loci (pQTL) datasets of plasma proteins to date and four large-scale GWAS summary statistics of lung function indices (FEV_1_, FVC, FEV1/FVC and peak expiratory flow [PEF]), to identify lung function indices-associated plasma proteins under a cutting-edge analytic framework by sequentially using PWAS, MR, and COLOC and mediation analysis. Specifically, we first performed PWAS to identify candidate proteins, followed by enrichment analysis to reveal the underlying biological processes and pathways. Then, we conducted MR analysis as well as COLOC analysis with discovery data to further select potentially causal proteins, followed by replicated MR analysis with replication data. Mediation analysis was also performed to explore if these lung function indices can serve as the possible mediators for the association between proteins and two common lung diseases, including chronic obstructive pulmonary disease (COPD) and asthma. Finally, we prioritized the potential drug targets.

## 2. Materials and Methods

The overall study design and comprehensive analysis procedure are displayed in Figure 1.

### 2.1. GWAS Data Source

We utilized the largest publicly available GWAS summary statistics with European ancestry to date for lung function indices and diseases. Specifically, the GWASs of FEV_1_, FVC and FEV_1_/FVC were undertaken with 321,047 individuals from UK Biobank and 79,055 individuals from the SpiroMeta Consortium. The GWAS of PEF was conducted with 321,047 individuals from UK Biobank and 24,218 individuals from the SpiroMeta Consortium [10]. The GWAS of COPD [22] (21,007 cases and 179,689 controls) and asthma [23] (56,087 cases and 428,511 controls) were both from the UK Biobank. All GWAS were approved by relevant ethic committees. Additional information including diagnostic criteria for cases, covariates adjusted during quality control were provided in Appendix A. Further genotyping methods and statistical analyses can be found in the original publications.

### 2.2. Human Plasma pQTL Data

The pQTL studies typically investigate the association between genetic variants and protein levels, which have recently been integrated with GWASs to elucidate the underlying mechanisms of complex traits [24]. In this study, two independent large-scale pQTL datasets with European ancestry were used as discovery and replication data sets, respectively. The relative concentrations of plasma proteins or protein complexes analyzed in both data were measured by Slow Off-Rate Modified Aptamers (SOMAmers) assay on the SomaLogic version-4 platform. The discovery dataset was derived from 7213 European Americans in the Atherosclerosis Risk in Communities (ARIC) study [25] including *cis*-pQTL data for 4657 plasma protein SOMAmers while the replication data were obtained from 35,559 Icelanders in deCODE genetics involving 27 million variants and 4907 plasma protein SOMAmers [26]. Both studies first regressed the plasma proteins on the covariates and obtain the residuals, and then performed the association analysis between the rank-inverse normalized residuals and genetic variants. Additional information including covariates adjusted during quality control were provided in Appendix A. Further genotyping methods and statistical analyses can be found in the original publications.

### 2.3. Statistical Analysis

#### 2.3.1. Proteome-Wide Association Study (PWAS)

PWAS aims to investigate the association between the genetically predicted proteins and complex traits through integrating the pQTL and GWAS. Typically, the z-score between protein and complex trait in PWAS was a linear sum of GWAS summary statistics multiplied by imputation weights of common variants at a certain locus, with the linkage disequilibrium (LD) accounted for. In this study, we conducted PWAS using FUSION pipeline [27] along with the predicted weights of 1348 *cis*-heritable proteins, which were identified to have significant nonzero *cis*-heritability (*p* < 0.01) from ARIC data. Trait specifically Bonferroni corrected *p*-value (0.05/1348) was used to control the multiple testing and declare the significance.

#### 2.3.2. Enrichment Analysis and Protein–Protein Interaction Network

To explore the biological functional interpretations of the proteins that are associated with the lung function indices from PWAS, we performed a gene set enrichment analysis using Metascape. Metascape implements enrichment analysis by identifying the gene sets whose members are significantly overrepresented using hypergeometric test and can eliminate confounding data interpretation issues by absorbing the majority of redundancies into representative clusters [28]. Here, we mainly focused on the biological process in Gene Ontology (GO) and Kyoto Encyclopedia of Genes and Genomes (KEGG). We performed the analysis with default parameter setting and kept the top 20 items for GO and top 10 for KEGG ranked by *p*-values. To delve further into the intricate web of interactions among the genes enriched in significant pathways from GO and KEGG, we employed protein–protein interaction (PPI) network analysis using STRING tool [29].

#### 2.3.3. Mendelian Randomization (MR) Analysis

For the significant protein–lung function index pairs identified by PWAS, we performed MR analyses, coupled with a series of sensitivity analyses, to evaluate the potential causal relationship between protein and lung function indices. For each pair, we first carried out discovery MR analysis using ARIC pQTL [25] data, and then conducted replication MR analysis using deCODE pQTL data [26]. Both MR analyses followed the strict procedure below and conformed to the STROBE-MR Statement [30], mainly involving the selection and assessment of F-statistics, primary MR analysis, and sensitivity analysis.

For instrumental variable (IV) selection, given that trans-pQTLs are likely to produce the pleiotropy issue and violate the basic assumption of MR analysis in assessing the potential causal effect of protein on the trait, we selected variants from the *cis*-region (500 kb upstream and downstream of the transcription start site of the protein-coding gene) of each protein. We adopted a less strict *p*-value threshold (*p* < 1 × 10^−5^) to determine significant pQTLs and obtained the independent *cis*-pQTLs (r^2^ < 0.01 in the 1 Mb *cis*-region) using the LD clumping procedure with the European LD reference panel in the 1000 Genomes Project. We further harmonized effect alleles of IVs in both exposure and outcome data. We calculated the F-statistic to evaluate the strength of instruments with values greater than 10 as evidence against weak instruments [31]. Further MR Steiger directionality test was applied to assess whether the MR analysis was biased by reverse causation.

For primary MR analysis, taking the different number of IVs for each protein into account, we used the Wald Ratio method for the proteins with only one IV, the fixed-effect inverse-variance weighted method (IVW) for proteins with two or three IVs, and the random-effect IVW method for proteins with four or more IVs. Of note, the random-effect model is able to account for heterogeneity across IVs by allowing for over-dispersion of the regression model. Further sensitivity analyses, including MR-Egger, leave-one-out and heterogeneity test, were also conducted to assess the robustness of the results. *p*-values were corrected using the Benjamani–Hochberg false discovery rate (BH-FDR) method, with FDR < 0.05 being declared to be significant. Similar MR analysis were further adopted to explore the causal relationship between potential causal proteins related to lung functions and common pulmonary diseases. These analyses were implemented using packages TwoSampleMR v.0.5.6, MendelianRandomization v.0.6.0 in the R v.4.2.1.

#### 2.3.4. Bayesian Colocalization Analysis

For the potential causal protein–trait pairs identified by MR analysis with discovery dataset, we further performed a colocalization analysis using the COLOC method [32] implemented in the R package COLOC (version 5.1.2) with the default setting. The colocalization analysis was able to investigate whether the association between protein and lung function indices was driven by common variants and help to examine the bias in MR analysis due to LD. The COLOC is a Bayesian method to assess the potential genetic patterns for each protein-trait pair with five hypotheses, of which hypothesis 4 (PP.H4) is our focus to reflect the setting that protein and trait share one causal SNP. We considered a significant colocalization signal with PP.H4 > 0.75. These analyses were implemented using packages COLOC v.5.2.0 in the R v.4.2.1.

#### 2.3.5. Mediation Analysis

To further explore whether these lung function indices could serve as the potential mediators for the effect of proteins on two common lung diseases, including COPD and asthma, we performed a mediation analysis under the MR framework with the product of coefficients method among proteins with potential causal effects on both lung functions and lung diseases. In the product method, the indirect effect could be estimated as the effect of proteins on the outcome through the mediator and confidence intervals were derived from the bootstrap approach. In addition, the mediation proportion will be calculated if the direction of the total effect and indirect effect are concordant [33].

#### 2.3.6. Candidate Druggable Targets

For proteins identified in the MR analysis and colocalization, we used Drug-Gene Interaction Database (DGIdb) (version 4.2.0) [34] and molecular docking [35] to explore whether they are the targets of the existing drugs or druggable gene targets, respectively. DGIdb integrates and normalizes information on drug-gene interactions from over 40 different resources. We first used it to prioritize drug target for disease. Subsequently, for the proteins that could interact with drugs from DGIdb, we further explored their molecular structure by molecular docking. The three-dimensional (3D) structures of drugs and proteins were obtained from PubChem (https://pubchem.ncbi.nlm.nih.gov/, accessd on 14 Febuary 2025), Protein Data Bank (http://www.rcsb.org/, accessed on 14 Febuary 2025), and AlphaFold (https://alphafold.com/, accessed on 14 Febuary 2025) predicted structures were used due to the lack of experimental data. Protein structures were preprocessed in PyMOL 2.4 by removing crystallographic water and adding polar hydrogens. AutoDock Tools 1.5.6 were used to prepare PDBQT files and define docking boxes. Molecular docking was performed using AutoDock Vina 1.2.2 (http://autodock.scripps.edu/, accessed on 16 Febuary 2025), with binding affinity assessed by binding free energy (ΔG, kcal/mol). Values below −7 kcal/mol indicated strong binding activity.

## 3. Results

### 3.1. PWAS Identified 210 Protein-Index Associations

By integrating the GWAS of four lung functions indices with the imputation of 1348 *cis*-heritable proteins from the ARIC pQTL data, we identified a total of 210 protein-index associations with Bonferroni-corrected *p*-value less than 0.05/1348, involving 130 unique proteins (Figure 2a with details are available in Appendix A). Among these 130 unique proteins, 10 proteins are associated with asthma, while 6 proteins are linked to COPD (Appendix A). Of note, 54 plasma proteins were related with two or more indices and 8 of them were simultaneously associated with FEV_1_, FVC, PEF and FEV_1_/FVC, suggesting the common pathogenesis of these four indices. For example, the Butyrophilin subfamily 3 member A3 encoded by gene *BTN3A3*, acting in T-cell responses in the adaptive immune response, was positively associated with these four indices (z-score > 0), while Complement C2 encoded by gene *C2*, as a part of pathway of the complement system, was negatively associated with them (z-score < 0). The enrichment analysis for the significant genes identified by PWAS revealed that GO terms relevant with pulmonary fibrosis and lung tissue repair were particularly prominent (Figure 2b), such as collagen-containing extracellular matrix (GO:0062023, *p* = 2.09 × 10^−19^) and morphogenesis of a branching epithelium (GO:0061138, *p* = 1.10 × 10^−10^). In addition, the proteins were mostly enriched in the pathways including PI3K-Akt signaling pathway (KEGG:hsa04151, *p* = 1.87× 10^−5^) and Wnt signaling pathway (KEGG:hsa04310, *p* = 7.54 × 10^−4^), which were well known to be related with the pathogenesis of lung function (Figure 2c). A total of 112 pathway-enriched proteins were subjected to PPI analysis. As shown in Figure 2d, we identified a significant network consisting of 66 enriched genes and 106 edges (*p* < 1 ×10^−16^), including the well-known lung function related genes *IL1B* [36,37], and *SHH* [38,39].

### 3.2. MR and COLOC Confirmed 59 Protein-Index Causal Relationships

For 210 protein-index pairs identified by PWAS, we first conducted MR discovery analysis using IVs derived from ARCI pQTL, to explore the potential causal relationship between the proteins and these indices. Initially, 231 instrumental variants for 130 proteins were extracted, among which 20 proteins had only one IV, 31 proteins had two or three IVs and 78 proteins had four or more IVs. Two pairs were excluded due to the lack of suitable IVs. F-statistics ranged from 19.8 to 595.8 (Appendix A), indicating less weak instrument bias. After harmonizing effect alleles, 208 plasma protein-index pairs were subjected to the MR analysis and several sensitivity analyses. MR-Steiger tests suggested that there was no reverse causal relationship, and MR-Egger did not show any horizontal pleiotropy (Appendix A). We finally obtained 175 significant protein-index pairs with FDR *p*-value < 0.05 (Appendix A).

Given that the false MR findings can be produced if the association between the protein and indices was driven by distinct variants with LD, we implemented COLOC for 175 protein-function pairs identified from MR analysis to remove such potentially false signals (Appendix A and Appendix A), and finally, 59 pairs showing significant evidence of colocalization remained, including 11 proteins for FEV_1_, 13 proteins for FVC, 14 proteins for PEF, and 21 proteins for FEV_1_/FVC (Table 1).

### 3.3. Forty-Two Protein-Index Causal Relationships Were Replicated

We further conducted replication MR analysis for the 58 protein-function pairs with deCODE pQTL data, with the remaining pair excluded due to the absence in the replication data. A total of 473 valid IVs were selected, which is 2.04 times higher than that from ARIC data, possibly due to the larger sample size. Following the same procedure above, 42 protein-function pairs were replicated with consistent effect estimates from discovery MR analysis, among which 23 pairs show positive associations and 19 pairs shows negative associations (Figure 3). Details are illustrated in Table 1, Appendix A and Appendix A.

### 3.4. Mediation Analysis Highlighted 3 Potential Pathways

For the 42 protein-index pairs obtained from both the discovery analysis and replication analysis, we further performed MR mediation analysis to assess the potential mediating roles of these lung function indices in the causal pathway from proteins to COPD or Asthma. Firstly, in MR analysis between proteins and pulmonary diseases, three proteins, including *EFEMP1*, *GM2A*, and *NPNT*, were identified that exhibit a causal relationship with COPD while five proteins were highlighted that have causal effect on asthma (Appendix A). Subsequently, with these proteins related to pulmonary diseases, we found three potential mediation pathways including indirect effect of *EFEMP1* on COPD via FEV_1_/FVC, an indirect effect of *MAPK3* on Asthma via FEV_1_/FVC and an indirect effect of *NPNT* on COPD risk via FEV_1_/FVC. The mediated proportions for these three pathways are 68.4%, 71.4%, and 82.7%, respectively (Figure 4).

### 3.5. Drug-Gene Interaction and Molecular Docking Discovery

Among the 34 unique protein-coding genes with significant evidence from both discovery and replication analysis, 24 were annotated as druggable genes (Appendix A) in DGIdb and 8 were reported to interact with drugs, including *FKBP4*, *FER*, *DENR*, *XPNPEP1*, *COL6A3*, *GM2A*, *MAPK3* and *SERPING1* (Appendix A). For the proteins encoded by above 8 genes interacting with drugs, we obtained 46 protein-drug pairs with 3D structures in molecular docking analysis. Finally, 3 proteins (*MAPK3*, *FER*, *XPNPEP1*) paired with 26 drugs were found with the docking criteria, where *MAPK3* exhibited a strong binding affinity with 15 drugs, *FER* interacted with 10 drugs, and *XPNPEP1* with 1 drug (Appendix A).

## 4. Discussion

In our study, through comprehensive and systematic omics integration analysis, we highlighted the role of plasma proteins in lung function and illuminated the potential mediating mechanism underlying the effect of proteins on common lung diseases. We have identified a total of 34 candidate proteins (24 druggable targets) with putatively causal effects on lung function, which could serve as novel therapeutic targets.

Some genes identified in this study, including *SFRP1*, *NPNT* and *EFEMP1*, were previously reported from the mouse model or allied experiments or association analysis, while the others were biologically plausible and are more likely to be vital for lung function. In particular, soluble frizzled-related protein-1, encoded by *SFRP1*, was demonstrated in the adult lung, leading to the destruction of lung tissue [40], and functioned as antagonizing the Wnt signaling which has a great impact in lung development and the progression of lung diseases including COPD, IPF and emphysema [41,42]. Several studies showed that the Nephronectin encoded by *NPNT* gene, as an extracellular protein, was highly expressed in human lung pneumocytes and alveolar fibroblasts cells [43,44] and necessary to maintain the normal development of lung [45,46] as well as contribute to COPD risk. In addition, the mRNA for the Fibulin-3 encoded by *EFEMP1* gene was discovered mostly abundant in the lung [47,48] in the mouse and associated with the incidence of restrictive physiology [49]. In addition, many significant genes were involved in several biological processes related to lung function disorder, for example, glutathione synthetase (*GSS*) was involved in the synthesis of glutathione (GSH), which could protect lung epithelium from damage in response to oxidants and inflammation [50]. Collagen Type VI Alpha 3 Chain (*COL6A3*) was confirmed to be involved in the construction of type VI collagen in most connective tissues [51,52].

Of note, plasma proteins identified by PWAS were significantly enriched in immune-related biological processes, which implied the close link between peripheral immunity and lung functions. Complex communications between immune system and respiratory disease have important etiological and clinical implications for lung disorders and have been of great interest in the last two decades. Some previous studies demonstrated that inflammation-sensitive plasma proteins were correlated with FVC or FEV1 [16,53]. Recent omics studies have provided a series of biomarkers of systemic inflammation reflecting important physiological roles in lung physiology, among which plasma C-reactive protein (CRP) may participate in the pathogenesis of various pulmonary disorders [54,55]. In addition, impaired pulmonary function resulted from immune disorders may play an important role in COPD and lung cancer [56,57].

Interestingly, we found that all significant proteins identified in the mediation analysis could serve as potential drug targets, with *MAPK3*-encoded protein being associated with 40 available drugs. Protein encoded by *MAPK3* was found to have an indirect effect on asthma mediated by FEV1/FVC. The MAPK/ERK pathway, including *MAPK3*, plays a role in the proliferation of airway smooth muscle cells, contributing to airway remodeling [58,59]. Thus, *MAPK3*-encoded protein may improve lung function and, in turn, ameliorate asthma. Similarly, both proteins encoded by *EFEMP1* and *NPNT* exhibited indirect effects on chronic obstructive pulmonary disease (COPD) by FEV1/FVC. These two proteins are involved in the regulating extracellular matrix (ECM), where its destruction and remodeling contribute to tissue damage and loss of airway elasticity [47,60]. Notably, the mediation proportions for these three proteins were 71.4%, 68.4%, and 82.7%, respectively, which indicated that there remained certain proportions of the protective effects of proteins not fully explained by lung function indices. These findings suggest that these proteins may exert protective effects independently of lung function, which is further supported by existing research.

One strength of our study was that we used two publicly available large-scale independent pQTL datasets of plasma protein and many GWASs of lung function indices, and implemented replication analysis to validate the finding. More importantly, other than focusing on the PWAS association, we sequentially used MR and COLOC analysis to further explore the protein-disorder causal associations to enhance the evidence of our findings. In addition, the mediation analysis further illustrated the potential mediating mechanism regarding the association between plasma protein and common lung diseases. However, our study is not without limitations. First, PWAS analysis was performed on only 1,348 proteins with significant *cis*-heritability due to the availability of genetic imputation model of protein expression; however, the remaining proteins may also play important roles in lung function. Second, all the analyses were based on European populations due to the data accessibility, and caution should be made when extending these findings to other populations. Third, due to limitations in accessing individual-level data of GWAS summary data, we were unable to consider the effects of medication when examining the two common chronic respiratory diseases. Fourth, all GWAS data on lung functions were derived from large populations, which include both healthy individuals and those with relevant diseases. The inclusion of disease conditions may introduce potential confounding, highlighting the importance of conducting analysis in different disease groups.

## 5. Conclusions

In summary, our findings provide further biological insights into the pathogenesis of impaired lung functions or common lung diseases through comprehensive analysis. We not only identified certain significant plasma proteins putatively causally associated with lung functions but also highlighted the potential mediating roles of lung function indices in the causal pathway from proteins to the common lung diseases. Meanwhile, these findings will benefit the development of preventive or therapeutic drugs for lung disorders, and explore the repurposing potential of existing drugs in the treatment of lung diseases.

## Figures and Tables

**Figure 1 cimb-47-00167-f001:**
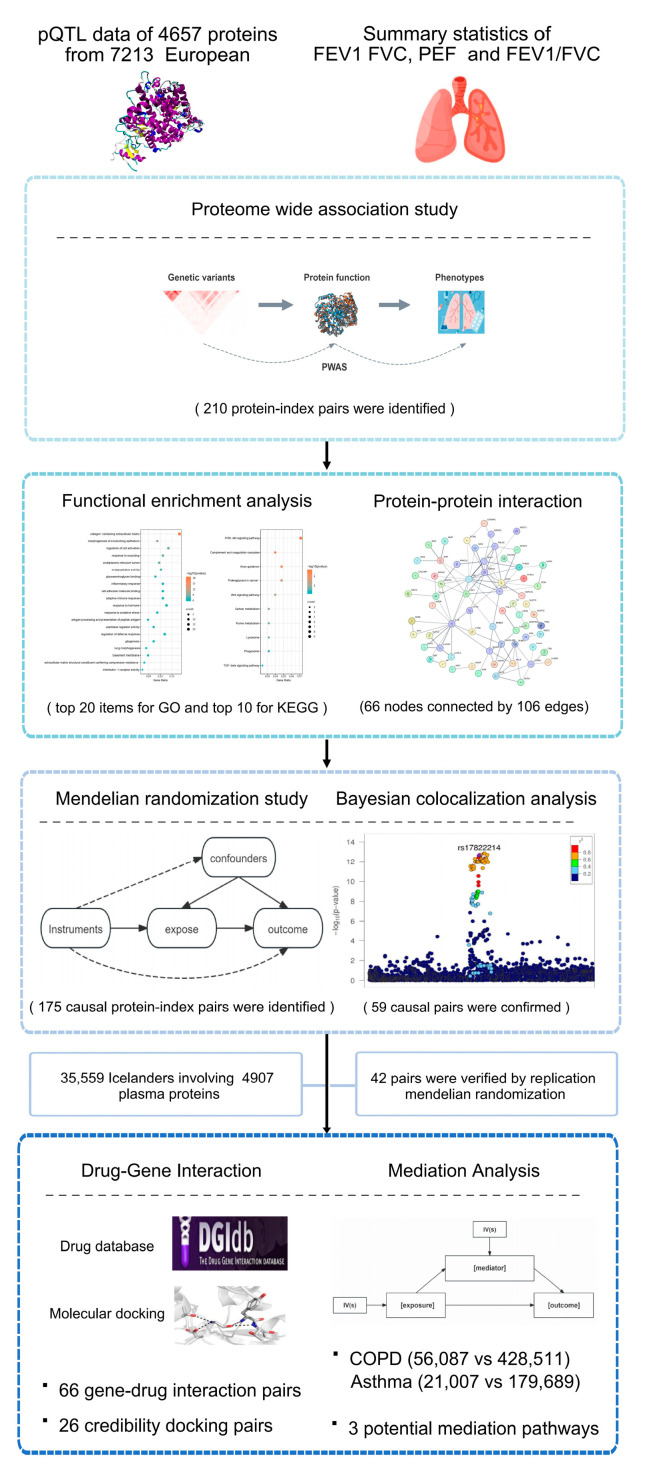
Study Workflow. We conducted a comprehensive omics integration analysis under a cutting-edge analytic framework by sequential using PWAS, MR, and COLOC to identify the plasma proteins associated with lung function indices and explore the potential mediating mechanism underlying the effect of proteins on lung diseases. We first identified protein-index associations by PWAS, followed by enrichment analysis and PPI analysis to reveal the underlying biological processes and pathways. Then, we performed MR and COLOC to further identify proteins causally associated with lung function indices followed by replication analysis. Mediation analysis was further performed to investigate the mediating role of these indices regarding the association between proteins and common lung disorders. Finally, we prioritized the potential drug targets. Abbreviations: pQTL, protein quantitative trait loci; GWAS, genome-wide association studies; PWAS, proteome-wide association study; GO, Gene Ontology; KEGG, Kyoto Encyclopedia of Genes and Genomes; MR mendelian randomization; FEV1, forced expiratory volume in 1 s; FVC, forced vital capacity; PEF, peak expiratory flow; COPD, chronic obstructive pulmonary disease; PPI, protein–protein interaction network.

**Figure 2 cimb-47-00167-f002:**
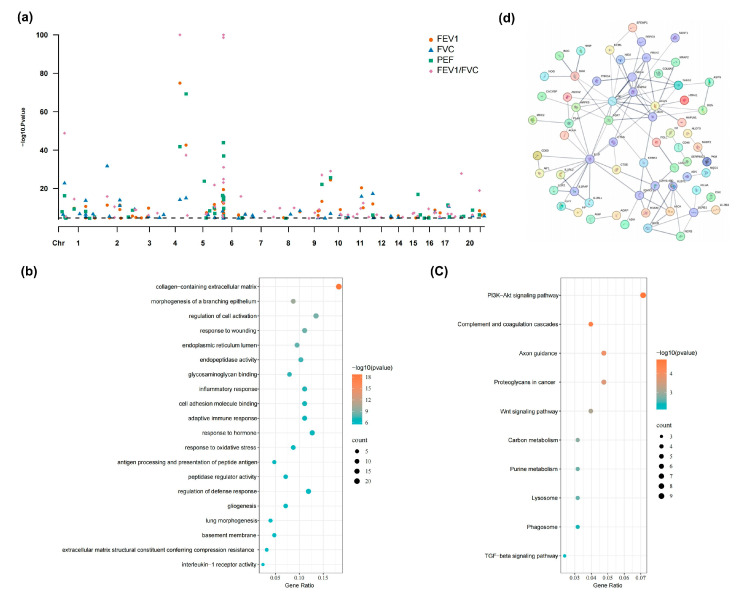
Results of PWAS and subsequent enrichment analysis. We identified a total of 210 protein-index associations and subsequently performed a gene set enrichment analysis using Metascape, with the top 20 significant GO terms or top 10 KEGG pathways displayed. (**a**) Manhattan plot for PWAS analysis of four lung function indices; (**b**) Bubble chart for GO enrichment analysis; (**c**) Bubble chart for KEGG enrichment analysis; (**d**) PPI with genes enriched in significant pathways. Abbreviations: FEV1, forced expiratory volume in 1 s; FVC, forced vital capacity; PEF, peak expiratory flow; PPI, Protein–protein interaction network.

**Figure 3 cimb-47-00167-f003:**
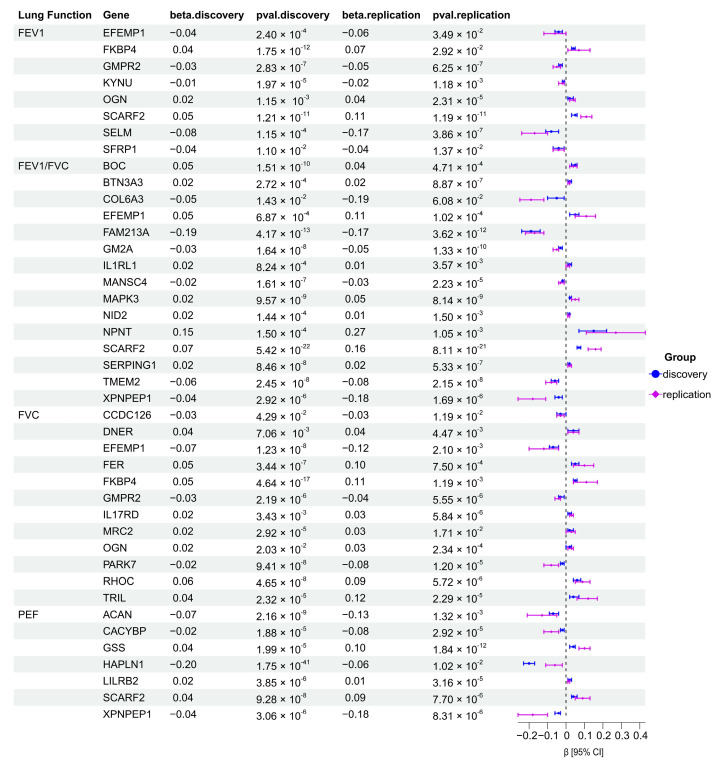
Results of the two-sample MR analysis. We used the ARIC data and deCODE data for discovery and replication analysis, respectively. The forest plot is utilized to illustrate the β values, *p*-values, and confidence intervals of mendelian randomization. FEV1, forced expiratory volume in 1 s; FVC, forced vital capacity; PEF, peak expiratory flow; IVW, inverse variance weighted method.

**Figure 4 cimb-47-00167-f004:**
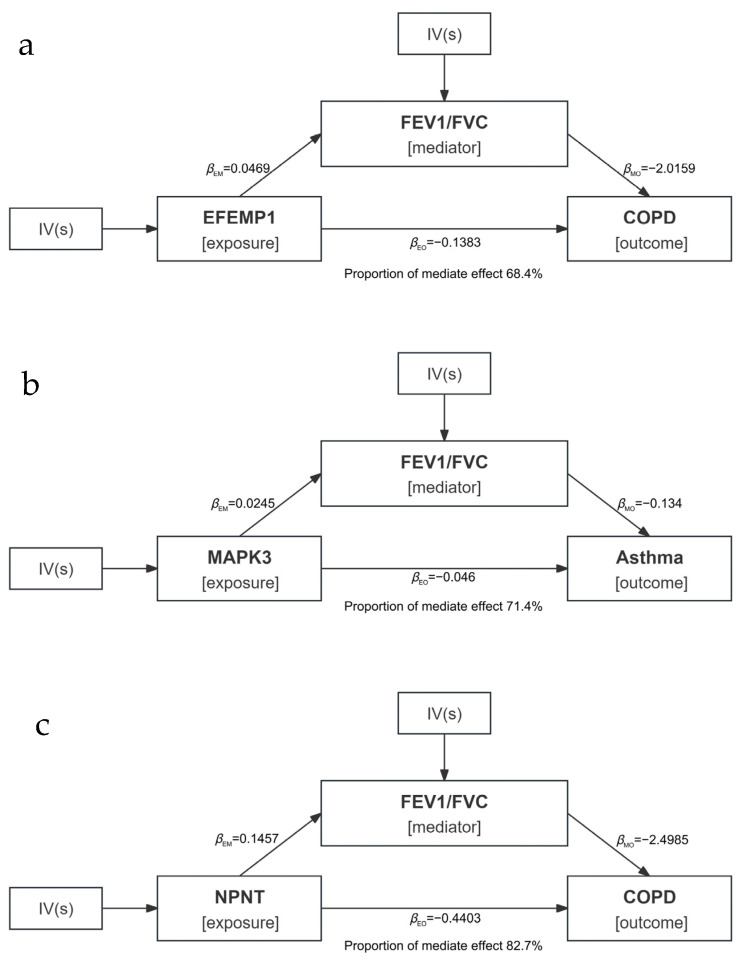
Mediation effects of protein on lung diseases via lung function indices. Mediation analyses to quantify the effects of three proteins on lung diseases via lung function indices. (**a**) EFEMP1 effect on COPD mediated by FEV1/FVC; (**b**) MARK3 effect on asthma mediated by FEV1/FVC; (**c**) NPNT effect on COPD mediated by FEV1/FVC. $\beta_{EM}$ represents the effect of exposure on mediator, $\beta_{MO}$ represents the effect of mediator on outcome, $\beta_{EO}$ represents the total effect of exposure on outcome. IV, instrumental variable; FEV1, forced expiratory volume in 1 s; FVC, forced vital capacity; COPD, chronic obstructive pulmonary disease.

**Table 1 cimb-47-00167-t001:** The 59 protein–lung function index pairs with high evidence of putatively causal relationship.

Lung Function Index	Gene	Chr	PWAS.P	Beta (95% CI)	PP.H4	Replicated	Existing Drug
FEV1	ASPN	9	1.12 × 10^−6^	−0.020 (−0.032, −0.008)	0.800	NO	NO
FEV1	EFEMP1	2	4.42 × 10^−12^	−0.039 (−0.060, −0.018)	0.976	Yes	NO
FEV1	FKBP4	12	1.12 × 10^−12^	0.041 (0.030, 0.053)	1.000	Yes	Yes
FEV1	FN1	2	2.22 × 10^−6^	0.044 (0.027, 0.061)	0.969	NO	NO
FEV1	GM2A	5	6.49 × 10^−10^	−0.023 (−0.034, −0.013)	0.981	NO	Yes
FEV1	GMPR2	14	1.64 × 10^−6^	−0.027 (−0.037, −0.017)	0.991	Yes	NO
FEV1	KYNU	2	3.06 × 10^−5^	−0.015 (−0.021, −0.008)	0.959	Yes	NO
FEV1	OGN	9	5.43 × 10^−9^	0.023 (0.009, 0.037)	0.978	Yes	NO
FEV1	SCARF2	22	3.60 × 10^−9^	0.047 (0.034, 0.061)	0.999	Yes	NO
FEV1	SELM	22	7.69 × 10^−7^	−0.076 (−0.114, −0.037)	0.963	Yes	NO
FEV1	SFRP1	8	1.55 × 10^−5^	−0.038 (−0.067, −0.009)	0.984	Yes	NO
FEV1/FVC	AGER	6	9.29 × 10^−173^	−0.186 (−0.199, −0.173)	1.000	NO	NO
FEV1/FVC	ARFIP1	4	1.02 × 10^−6^	0.027 (0.016, 0.038)	0.976	NO	NO
FEV1/FVC	BOC	3	6.47 × 10^−8^	0.049 (0.034, 0.063)	0.988	Yes	NO
FEV1/FVC	BTN3A3	6	1.28 × 10^−6^	0.016 (0.008, 0.025)	0.859	Yes	NO
FEV1/FVC	COL6A3	2	1.39 × 10^−9^	−0.053 (−0.096, −0.011)	0.999	Yes	Yes
FEV1/FVC	EFEMP1	2	1.41 × 10^−16^	0.047 (0.020, 0.074)	0.891	Yes	NO
FEV1/FVC	FAM213A	10	6.04 ×10^−7^	−0.187 (−0.238, −0.136)	0.983	Yes	NO
FEV1/FVC	GM2A	5	4.37 × 10^−11^	−0.027 (−0.036, −0.018)	0.770	Yes	Yes
FEV1/FVC	HP	16	2.04 × 10^−8^	0.021 (0.013, 0.029)	0.973	NO	NO
FEV1/FVC	IL1RL1	2	3.28 × 10^−11^	0.017 (0.007, 0.028)	0.999	Yes	NO
FEV1/FVC	LTBP4	19	1.53 × 10^−28^	−0.197 (−0.229, −0.165)	1.0000	NO	NO
FEV1/FVC	MANSC4	12	4.84 × 10^−7^	−0.016 (−0.022, −0.010)	0.856	Yes	NO
FEV1/FVC	MAPK3	16	3.93 × 10^−10^	0.025 (0.016, 0.033)	0.999	Yes	Yes
FEV1/FVC	NID2	14	8.49 × 10^−6^	0.016 (0.008, 0.024)	0.868	Yes	NO
FEV1/FVC	NOG	17	1.80 × 10^−7^	0.064 (0.032, 0.096)	0.995	NO	NO
FEV1/FVC	NPNT	4	7.84 × 10^−116^	0.146 (0.070, 0.221)	1.000	Yes	NO
FEV1/FVC	PAPPA	9	9.05 × 10^−28^	0.350 (0.287, 0.412)	0.993	NO	NO
FEV1/FVC	SCARF2	22	1.19 × 10^−19^	0.069 (0.055, 0.083)	0.999	Yes	NO
FEV1/FVC	SERPING1	11	2.32 × 10^−10^	0.017 (0.010, 0.023)	0.996	Yes	Yes
FEV1/FVC	TMEM2	9	4.67 × 10^−8^	−0.061 (−0.082, −0.039)	0.998	Yes	NO
FEV1/FVC	XPNPEP1	10	1.32 × 10^−5^	−0.041 (−0.058, −0.024)	0.933	Yes	Yes
FVC	CA3	8	3.02 × 10^−7^	−0.046 (−0.062, −0.029)	0.997	NO	NO
FVC	CCDC126	7	1.14 × 10^−6^	−0.026 (−0.052, −0.001)	0.880	Yes	NO
FVC	DNER	2	7.47 × 10^−10^	0.038 (0.011, 0.066)	0.998	Yes	Yes
FVC	EFEMP1	2	2.16 × 10^−32^	−0.066 (−0.089, −0.044)	0.999	Yes	NO
FVC	FER	5	5.67 × 10^−7^	0.051 (0.032, 0.071)	0.997	Yes	Yes
FVC	FKBP4	12	5.24 × 10^−18^	0.049 (0.038, 0.061)	1.000	Yes	Yes
FVC	GMPR2	14	1.51 × 10^−5^	−0.025 (−0.036, −0.015)	0.963	Yes	NO
FVC	IL17RD	3	4.71 × 10^−6^	0.017 (0.006, 0.029)	0.820	Yes	NO
FVC	MRC2	17	2.51 × 10^−11^	0.025 (0.013, 0.036)	0.956	Yes	NO
FVC	OGN	9	2.17 × 10^−5^	0.017 (0.003, 0.031)	0.793	Yes	NO
FVC	PARK7	1	3.87 × 10^−7^	−0.019 (−0.027, −0.012)	0.978	Yes	NO
FVC	RHOC	1	1.83 × 10^−7^	0.056 (0.036, 0.076)	0.989	Yes	NO
FVC	TRIL	7	9.55 × 10^−6^	0.045 (0.024, 0.066)	0.980	Yes	NO
PEF	ACAN	15	4.95 × 10^−7^	−0.066 (−0.088, −0.044)	1.000	Yes	NO
PEF	AGER	6	1.13 × 10^−37^	−0.091 (−0.105, −0.078)	0.999	NO	NO
PEF	CACYBP	2	2.98 × 10^−5^	−0.023 (−0.034, −0.013)	0.944	Yes	NO
PEF	FKBP4	12	6.54 × 10^−7^	0.030 (0.016, 0.044)	0.998	NO	Yes
PEF	FN1	2	2.27 × 10^−9^	0.053 (0.035, 0.072)	0.996	NO	NO
PEF	GM2A	5	9.64 × 10^−8^	−0.022 (−0.032, −0.013)	0.951	NO	Yes
PEF	GSS	20	1.66 × 10^−9^	0.037 (0.020, 0.055)	0.886	Yes	NO
PEF	HAPLN1	5	1.44 × 10^−24^	−0.202 (−0.232, −0.173)	0.976	Yes	NO
PEF	LILRB2	19	2.79 × 10^−6^	0.018 (0.011, 0.026)	0.998	Yes	NO
PEF	LTBP4	19	8.36 × 10^−6^	−0.093 (−0.127, −0.059)	1.000	NO	NO
PEF	NOG	17	4.18 × 10^−12^	0.102 (0.068, 0.136)	0.959	NO	NO
PEF	NPNT	4	1.55 × 10^−42^	0.092 (0.044, 0.140)	1.000	NO	NO
PEF	SCARF2	22	4.88 × 10^−7^	0.041 (0.026, 0.056)	0.995	Yes	NO
PEF	XPNPEP1	10	1.59 × 10^−5^	−0.043 (−0.062, −0.025)	0.924	Yes	Yes

The estimate of causal effects (beta) from MR analysis and the posterior probability (PP.H4) from COLOC analysis with the discovery samples for the final 59 protein-function pairs. Pairs identified in the replication analysis are annotated as Yes in the Replicated column. Genes that are interacted with existing drugs in Drug Gene Interaction Database are annotated as Yes in the Existing drug column. Chr = Chromosome. PWAS.P = *p*-value from PWAS analysis. CI = Confidence interval. FEV1 = Forced expiratory volume in 1 s. FVC = Forced vital capacity. PEF = Peak expiratory flow. MR = Mendelian Randomization. PWAS = Proteome-wide association study.

## Data Availability

The datasets generated and/or analyzed during the current study are publicly available. GWAS summary data for FVC, FEV1, FEV1/FVC, PEF, Asthma and COPD are available from GWAS Catalog (study accession: GCST007429, GCST007432, GCST007431, GCST007430, GCST90038616, GCST90016586) at https://www.ebi.ac.uk/gwas/ (accessed on 29 December 2024). The two pQTL data are available from Atherosclerosis Risk in Communities study at http://nilanjanchatterjeelab.org/pwas/ (accessed on 29 December 2024) and deCODE genetics at https://www.decode.com/summarydata/ (accessed on 29 December 2024).

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
