# Peer review of "Identification of Candidate Lung Function-Related Plasma Proteins to Pinpoint Drug Targets for Common Pulmonary Diseases: A Comprehensive Multi-Omics Integration Analysis"

_cimb, 2025, doi:10.3390/cimb47030167_

Round 1
Reviewer 1 Report
Comments and Suggestions for Authors
In this study, the authors conducted a integration analysis to identified candidate lung function-related plasma proteins to pinpoint drug targets for common pulmonary diseases. Generally, this paper is well prepared. Nevertheless, the overall quality may be further improved with careful revision of the following issues.
Q1, Were downstream pathway of action analyses for gene enrichment analyses performed?
Q2, Are there any attempts to use techniques such as molecular docking for further analysis in the drug candidate screening section?
Q3, The authors performed multiple bioinformatics analyses using FEV1, FVC, FEV1/FVC, and PEF as outcome variables, but these metrics confounded healthy individuals and patients with different lung diseases. Are the results clinically meaningful?
Q4, Have subgroup analyses been attempted to clarify plasma protein phenotypic changes in different lung diseases such as COPD, asthma and pneumonia?
Q5, The writing of the manuscript needs improvement with the assistance of English editing.
Comments on the Quality of English LanguageThe writing of the manuscript needs improvement with the assistance of English editing.
Author Response
In this study, the authors conducted an integration analysis to identified candidate lung function-related plasma proteins to pinpoint drug targets for common pulmonary diseases. Generally, this paper is well prepared. Nevertheless, the overall quality may be further improved with careful revision of the following issues.
Responses: Thanks very much for your positive review and constrictive comments, which have led to great improvement of our manuscript. We have taken care to address your concerns in the revised manuscript, with detailed point-by-point responses to each of your specific comments provided below.
Q1, Were downstream pathway of action analyses for gene enrichment analyses performed?
Responses: Thanks for your suggestion. Following your comment, targeting on the 112 genes significantly enriched in the pathways from gene enrichment analysis, we further conducted protein–protein interaction network (PPI) analysis to investigate the potentially biological mechanisms1,2 by the STRING tool3. As shown in Figure below, we found a significant PPI network consisting of 66 genes and 106 edges (P < 1e-16), including the well-known lung function related genes IL1B5,6, and SHH7,8. We have added these results in the revised manuscript (the first paragraph in Page 5, the first paragraph in Page 7, updated Figure 1, and updated Figure 2)
Figure Protein–protein interaction network with genes enriched in significant pathways
Q2, Are there any attempts to use techniques such as molecular docking for further analysis in the drug candidate screening section?
Responses: Thanks for your valuable comments. Based on your suggestions, we further adopted molecular docking analysis on the 8 proteins interacted with drugs in DGIdb, leading to a total of 46 protein-drug pairs, to valid the druggability from a molecular mechanism perspective. Finally, 3 proteins (MAPK3, FER, XPNPEP1) paired with 26 drugs were found with the docking criteria (Energy value < -7 kcal/mol), where MAPK3 exhibited a strong binding affinity with 15 drugs, FER interacted with 10 drugs, and XPNPEP1 with 1 drug. We have added these detailed results in the revised manuscript (see the third paragraph in Page 6, the last paragraph in Page 11, updated Figure 1, and Supplementary table S13)
Q3, The authors performed multiple bioinformatics analyses using FEV1, FVC, FEV1/FVC, and PEF as outcome variables, but these metrics confounded healthy individuals and patients with different lung diseases. Are the results clinically meaningful?
Responses: Thanks for your comment. We speculated that your concern was mainly focused on the confounding effect of the disease status on the association between proteins and lung function, and stratified analysis should be conducted in individuals with different lung disease groups. In the manuscript, we performed the analysis using large-scale publicly available GWAS summary statistics for lung function. Following your comment, we have searched the related literature and found that all existed GWAS for lung function involved both healthy individuals and those with relevant diseases9,10. In addition, due to privacy and ethical policy, we are unable to access the individual data and failed to conduct the subsequent subgroup analysis, such as performing the omics integration analysis in healthy individuals and patients with different lung diseases, respectively. Even we have included Mendelian Randomization (MR) analysis, which is well-known to be less susceptible to confounding11, in the comprehensive analytical pipeline to alleviate this issue. We, following your instructions, still discussed this as a limitation in the revised manuscript to highlight the importance of conducting analysis in different disease groups. (the second paragraph on Page 13)
Q4, Have subgroup analyses been attempted to clarify plasma protein phenotypic changes in different lung diseases such as COPD, asthma and pneumonia?
Responses: Thanks for your valuable suggestions. To clarify plasma protein phenotypic changes in different lung diseases, we, following your comments, conducted the PWAS analysis to explore the relationship between plasma proteins and COPD, as well as between plasma proteins and asthma, on the 130 unique proteins associated with lung functions, where 6 and 10 proteins were identified to be associated with COPD and asthma, respectively. In addition, we also performed the MR analysis to investigate the causal relationship between plasma proteins and COPD, as well as between plasma proteins and asthma, on the 34 unique proteins causally associated with lung functions, where 3 and 5 proteins were identified to have the potential causal effects on COPD and asthma, respectively. We finally conducted mediation analysis to investigate the underlying mechanisms of these potential causal proteins. All these results are added in the revised manuscript (the last paragraph in Page 6, the last paragraph in Page 10, Supplementary Table S3, S9 and S10).
Q5, The writing of the manuscript needs improvement with the assistance of English editing.
Responses: Thanks for your kind reminder. We have asked a native English speaker for help to edit the language and make the manuscript more concise. We have also carefully checked the manuscript again to minimize the grammar and spelling error, to improve the readability.
Reference
1. Deng YT, Ou YN, Wu BS, et al. Identifying causal genes for depression via integration of the proteome and transcriptome from brain and blood. Mol Psychiatry 2022; 27(6): 2849-57.
2. Wen S, Xu S, Zong X, et al. Association Analysis of the Circulating Proteome With Sarcopenia-Related Traits Reveals Potential Drug Targets for Sarcopenia. J Cachexia Sarcopenia Muscle 2025; 16(1): e13720.
3. Szklarczyk D, Kirsch R, Koutrouli M, et al. The STRING database in 2023: protein-protein association networks and functional enrichment analyses for any sequenced genome of interest. Nucleic Acids Res 2023; 51(D1): D638-D46.
4. Tabei Y, Nakajima Y. IL-1beta-activated PI3K/AKT and MEK/ERK pathways coordinately promote induction of partial epithelial-mesenchymal transition. Cell Commun Signal 2024; 22(1): 392.
5. Zhang S, Fan Y, Qin L, et al. IL-1beta augments TGF-beta inducing epithelial-mesenchymal transition of epithelial cells and associates with poor pulmonary function improvement in neutrophilic asthmatics. Respir Res 2021; 22(1): 216.
6. Hu Y, Peng L, Zhuo X, Yang C, Zhang Y. Hedgehog Signaling Pathway in Fibrosis and Targeted Therapies. Biomolecules 2024; 14(12).
7. Zeng LH, Barkat MQ, Syed SK, et al. Hedgehog Signaling: Linking Embryonic Lung Development and Asthmatic Airway Remodeling. Cells-Basel 2022; 11(11).
8. Chen CY, Chen TT, Feng YA, et al. Analysis across Taiwan Biobank, Biobank Japan, and UK Biobank identifies hundreds of novel loci for 36 quantitative traits. Cell Genom 2023; 3(12): 100436.
9. Shrine N, Izquierdo AG, Chen J, et al. Multi-ancestry genome-wide association analyses improve resolution of genes and pathways influencing lung function and chronic obstructive pulmonary disease risk. Nat Genet 2023; 55(3): 410-22.
10. Burgess S, Mason AM, Grant AJ, et al. Using genetic association data to guide drug discovery and development: Review of methods and applications. Am J Hum Genet 2023; 110(2): 195-214.

Reviewer 2 Report
Comments and Suggestions for Authors
Summary:
This manuscript reports a work aimed to identify target molecular functions and proteins associated with lung diseases as indicated by lung function indices like FEV1, FVC, FEV1/FVC, PEF. This work starts with datasets of Genome-wide association studies reporting associations between genomic variants and measured lung function indices utilizing an integrative omics approach. In order to narrow down the search space and causal associations a pathways and molecular level. They performed proteome-wide association analysis followed by Mendelian-randomization and Bayesian colocalization analysis. Two common lung diseases Asthma and chronic obstructive pulmonary disease and their association to lung function indices is also performed to investigate the indirect roles of proteins to these diseases. The analysis identified 59 causal protein-index pairs, among which 42 pairs were replicated. Further they studied the druggability of identified and found a total of 24 proteins as druggable targets in Drug Gene Interaction Database, including FKBP4, GM2A, COL6A3, MAPK3, SERPING1, XPNPEP1, DNER, and FER, that are reported to interact with drugs in the database.
The manuscript presents a currently relevant and important work which may aid in establishing target pathways and proteins after experimental validations related to lung diseases and upon validation may be used for further drug-discovery efforts related to such diseases.
The manuscript clearly underscores the need of the work, outlines a clear study plan and presents the results concisely and discusses the relevance of the outcomes.
Majors.
1. Authors have used both “forced expired volume in 1 second” and “forced expiratory volume in 1 second”, however, I suggest to use the commonly used term “ forced expiratory volume in 1 second” consistently.
2. Authors state in introduction paragraph 2 “Although some studies have reported the associations between plasma proteins and lung functions, the scales of proteins included in these analyses are limited compared to whole plasma proteome.”, however, they failed to provide the references to such studies.
3. At the end of Materials and Methods authors have listed the abbreviations, which are generally provided at the end of the manuscript under appropriate heading.
4. At the end of section “GWAS data source” authors state that “Additional information including diagnostic criteria for cases, genotyping method, quality control and statistical analyses, were described in the original publications.” though it refers to how these were done in the cited works. Authors should add a brief summary of these to navigate readers to how to interpret the results and discussion in following sections and to what extent. Same applies to “The detailed quality control was described in the original article.” in the section “Human plasma pQTL data”.
Minors.
1. In “mainly involving the selection and assessment of f, primary MR analysis as well as sensitivity analysis.” Authors should correct “assessment of f” to “assessment of F-statistics”.
2. Figure 2, (a) the colors used for lung function indices are hard to distinguish, authors may consider using different markers for each test to enhance the readability of the figure. The x-axis labels in both (b) and (c) are missing.
3. Sentence “F-statistics for were ranged from 19.8 to 595.8” is grammatically incorrect, word “were” is unnecessary.
4. “We finally remained 59 pairs showing significant evidence of colocalization”, here “We” is not needed.
5. Figure 3, x-axis label is missing.
6. References has double numbers.
Comments on the Quality of English LanguageThe overall English of the manuscript is fine, except for a few corrections that are needed.
Author Response
This manuscript reports a work aimed to identify target molecular functions and proteins associated with lung diseases as indicated by lung function indices like FEV1, FVC, FEV1/FVC, PEF. This work starts with datasets of Genome-wide association studies reporting associations between genomic variants and measured lung function indices utilizing an integrative omics approach. In order to narrow down the search space and causal associations a pathways and molecular level. They performed proteome-wide association analysis followed by Mendelian-randomization and Bayesian colocalization analysis. Two common lung diseases Asthma and chronic obstructive pulmonary disease and their association to lung function indices is also performed to investigate the indirect roles of proteins to these diseases. The analysis identified 59 causal protein-index pairs, among which 42 pairs were replicated. Further they studied the druggability of identified and found a total of 24 proteins as druggable targets in Drug Gene Interaction Database, including FKBP4, GM2A, COL6A3, MAPK3, SERPING1, XPNPEP1, DNER, and FER, that are reported to interact with drugs in the database.
The manuscript presents a currently relevant and important work which may aid in establishing target pathways and proteins after experimental validations related to lung diseases and upon validation may be used for further drug-discovery efforts related to such diseases.
The manuscript clearly underscores the need of the work, outlines a clear study plan and presents the results concisely and discusses the relevance of the outcomes.
Responses: Thanks for your positive review and acknowledgements with our manuscript. Your constrictive comments have led to great improvement of our manuscript. Detailed point-by-point responses to each of your specific comments are provided below.
Majors.
1. Authors have used both “forced expired volume in 1 second” and “forced expiratory volume in 1 second”, however, I suggest to use the commonly used term “ forced expiratory volume in 1 second” consistently.
Responses: Thanks for pointing this out. We have unified the term through the whole manuscript to be ‘forced expiratory volume in 1 second’ as you suggested.
2. Authors state in introduction paragraph 2 “Although some studies have reported the associations between plasma proteins and lung functions, the scales of proteins included in these analyses are limited compared to whole plasma proteome.”, however, they failed to provide the references to such studies.
Responses: Thanks a lot. We have added the references and clarified that current studies on the relationship between plasma proteins and lung function primarily focus on immune-related proteins, with limited the exploration of the whole plasma proteome.1-4 (see the second paragraph in Page 2)
3. At the end of Materials and Methods authors have listed the abbreviations, which are generally provided at the end of the manuscript under appropriate heading.
Responses: Thanks for your comment. We are speculated that the abbreviation list you referred here is the one included in the legend of Figure 1, which is quite necessary to navigate the readers better to capture the main message convey by the comprehensive analysis pipeline. Thus, we chose to retain them in the legends of Figure 1.
4. At the end of section “GWAS data source” authors state that “Additional information including diagnostic criteria for cases, genotyping method, quality control and statistical analyses, were described in the original publications.” though it refers to how these were done in the cited works. Authors should add a brief summary of these to navigate readers to how to interpret the results and discussion in following sections and to what extent. Same applies to “The detailed quality control was described in the original article.” in the section “Human plasma pQTL data”.
Responses: Thanks for your comment. In the original manuscript, we have included the details of data source in the in Supplementary Table 1, including the diagnostic criteria and covariates adjusted during quality control. We have involved a total of 6 large-scale GWAS studies and 2 pQTL studies, and different studies have different quality control and statistical analyses. Therefore, following such similar omics integration analysis works5-7, we did not summarize additional specific details in the main text per the page limit, while chose to refer the original publications. (see the second and the third paragraph in Page 4, and Supplementary Table 1)
Minors.
1. In “mainly involving the selection and assessment of f, primary MR analysis as well as sensitivity analysis.” Authors should correct “assessment of f” to “assessment of F-statistics”.
Responses: Thanks for your kind reminder. We have corrected this statement in the revised manuscript. (see the third paragraph in Page 5)
2. Figure 2, (a) the colors used for lung function indices are hard to distinguish, authors may consider using different markers for each test to enhance the readability of the figure. The x-axis labels in both (b) and (c) are missing.
Responses: Thanks for pointing this out. we have updated Figure 2 with different colors and distinct shapes (orange circle for FEV1, blue triangle for FVC, green square for PEF, pink diamond for FEV1/FVC.) to represent various lung functions, to enhance the readability. In addition, we have included the x-axis in panels b and c (the updated Figure 2).
3. Sentence “F-statistics for were ranged from 19.8 to 595.8” is grammatically incorrect, word “were” is unnecessary.
Responses: We have corrected this (the first paragraph in Page 7).
4. “We finally remained 59 pairs showing significant evidence of colocalization”, here “We” is not needed.
Responses: We have corrected this (see the second paragraph in Page 8).
5. Figure 3, x-axis label is missing.
Responses: We have added the x-axis label in Figure 3 (the updated Figure 3).
6. References has double numbers.
Responses: We have modified this (see the References).
Reference
1. Engstrom G, Lind P, Hedblad B, et al. Lung function and cardiovascular risk: relationship with inflammation-sensitive plasma proteins. Circulation 2002; 106(20): 2555-60.
2. Dahl M, Tybjaerg-Hansen A, Vestbo J, Lange P, Nordestgaard BG. Elevated plasma fibrinogen associated with reduced pulmonary function and increased risk of chronic obstructive pulmonary disease. Am J Respir Crit Care Med 2001; 164(6): 1008-11.
3. Sunyer J, Pistelli R, Plana E, et al. Systemic inflammation, genetic susceptibility and lung function. Eur Respir J 2008; 32(1): 92-7.
4. Keefe J, Yao C, Hwang SJ, et al. An Integrative Genomic Strategy Identifies sRAGE as a Causal and Protective Biomarker of Lung Function. Chest 2022; 161(1): 76-84.
5. Zhang Y, Yu S, Chen Z, et al. Gestational diabetes and future cardiovascular diseases: associations by sex-specific genetic data. Eur Heart J 2024; 45(48): 5156-67.
6. Li Z, Zhang B, Liu Q, et al. Genetic association of lipids and lipid-lowering drug target genes with non-alcoholic fatty liver disease. EBioMedicine 2023; 90: 104543.
7. Chong M, Sjaarda J, Pigeyre M, et al. Novel Drug Targets for Ischemic Stroke Identified Through Mendelian Randomization Analysis of the Blood Proteome. Circulation 2019; 140(10): 819-30.
